# UFC-BERT: Unifying Multi-Modal Controls for Conditional Image Synthesis

**Zhu Zhang**[†], **Jianxin Ma**[†], **Chang Zhou**[†], **Rui Men**[†], **Zhikang Li**[†],
**Ming Ding**[‡], **Jie Tang**[‡], **Jingren Zhou**[†], **and Hongxia Yang**[†]
[†]DAMO Academy, Alibaba Group, [‡]Tsinghua University
{zhangzhu950310}@gmail.com
{jason.mjx, ericzhou.zc, yang.yhx}@alibaba-inc.com

## Abstract

Conditional image synthesis aims to create an image according to some multi-modal guidance in the forms of textual descriptions, reference images, and image blocks to preserve, as well as their combinations. In this paper, instead of investigating these control signals separately, we propose a new two-stage architecture, UFC-BERT, to unify any number of multi-modal controls. In UFC-BERT, both the diverse control signals and the synthesized image are uniformly represented as a sequence of discrete tokens to be processed by Transformer. Different from existing two-stage autoregressive approaches such as DALL-E and VQGAN, UFC-BERT adopts non-autoregressive generation (NAR) at the second stage to enhance the holistic consistency of the synthesized image, to support preserving specified image blocks, and to improve the synthesis speed. Further, we design a progressive algorithm that iteratively improves the non-autoregressively generated image, with the help of two estimators developed for evaluating the compliance with the controls and evaluating the fidelity of the synthesized image, respectively. Extensive experiments on a newly collected large-scale clothing dataset M2C-Fashion and a facial dataset Multi-Modal CelebA-HQ verify that UFC-BERT can synthesize high-fidelity images that comply with flexible multi-modal controls.

## 1 Introduction

Conditional image synthesis aims to create an image according to the given control signals. With the increasing demand for flexible conditional image synthesis, various kinds of control signals have been introduced into this field, which can be divided into three main modalities: (i) *textual controls (TC)*, including the class labels [1] and natural language descriptions [62, 54]; (ii) *visual controls (VC)*, such as a spatially-aligned sketch map for reference [17, 60] or another image for style transfer [15, 27]; (iii) *preservation controls (PC)*, which require the synthesized image to preserve some given image blocks, e.g., image outpainting and inpainting [63, 69].

However, control signals of various modalities possess different characteristics. Existing works [62, 26, 61] hence typically design separate methods customized for each control modality. Moreover, most of these approaches only utilize one type of control signal and cannot simultaneously combine multiple types of controls in a concise and versatile model. This begs the question: can we integrate any number of multi-modal control signals into a unified framework for flexible conditional image synthesis? There are two inevitable challenges in this setting: (i) how to unify the multi-modal controls and represent them in a unified form, especial when employing multiple control signals from different modalities concurrently; (ii) how to guarantee the fulfillment of the multi-modal controls while ensuring the fidelity of the synthesized image.

35th Conference on Neural Information Processing Systems (NeurIPS 2021).

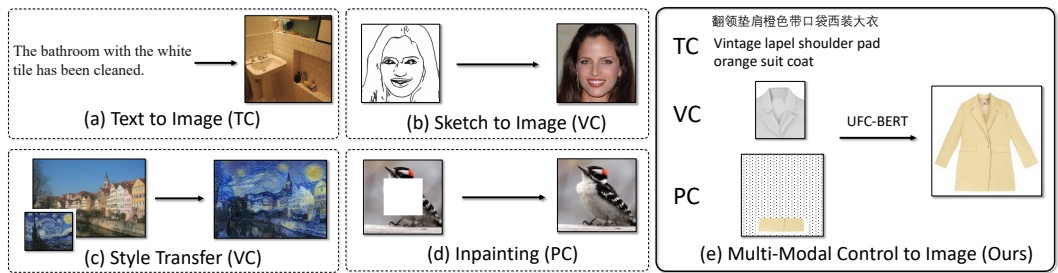

Figure 1: The three main modalities of control signals for conditional image synthesis: Textual Controls (TC), Visual Controls (VC), and Preservation Controls (PC).

Recently, two-stage image synthesis [42, 48, 3, 13, 47] has made great progress. The first stage learns a convolutional autoencoder with quantized latent representations for converting an image into a sequence of discrete tokens, e.g., for compressing a 256×256 image into a sequence of $32 \times 32$ tokens where each token correlates mainly with an $8 \times 8$ block of the image. Converting a sequence of tokens back into an image is also supported. The second stage then typically adopts an autoregressive model, e.g., PixelCNN [41] or a unidirectional Transformer decoder [55], to capture the distribution over sequences of tokens. Particularly, the Transformer-based methods [3, 13, 47] exploit the global expressivity of Transformer to capture long-range relationships between local constituents.

In this paper, we make two key observations about the two-stage framework. First, the two-stage framework has the advantage that it can potentially unify the multi-modal control signals and the generated image into a single sequence of discrete tokens. However, existing works [42, 48, 13, 47] largely neglect this advantage of the two-stage framework over the traditional one-stage approaches such as those based mainly on the generative adversarial networks (GAN) [18]. Second, the *autoregressive (AR)* approach to sequence generation, adopted by the existing two-stage methods such as DALL-E [47] and VQGAN [13], brings undesirable shortcomings: (i) the token-by-token synthesis procedure leads to slow generation speed, especially for the heavyweight Transformer [3, 13, 47]; (ii) each generated token can only catch sight of the previously generated tokens and cannot incorporate bidirectional contexts, which may affect the holistic consistency of image synthesis; (iii) the fixed left-to-right order of autoregressive decoding cannot respond to the preservation control signals unless the image blocks to be preserved are at the beginning of the sequence. Notably, different from AR generation, *non-autoregressive (NAR)* sequence generation with bidirectional Transformer, i.e., BERT [9], can naturally avoid the three shortcomings.

Based on the aforementioned observations, we propose UFC-BERT, a novel BERT-based two-stage framework to **U**ni**F**y any number of multi-modal **C**ontrols for conditional image synthesis. Concretely, the textual, visual, and preservation control signals, as well as the generated image, are uniformly represented as a sequence of discrete tokens, as shown in Figure 2. The textual control consists of word tokens for class labels or natural language descriptions. The visual control(s) and the generated image are both represented as discrete tokens due to the first stage, where each token corresponds to a block within the reference image(s) or the generated image. Zero, one, or more reference images are supported. To preserve a given image block within the generated image, we encode the given image block into discrete tokens and fix corresponding parts of the generated sequence to the tokens.

We train UFC-BERT via the masked sequence modeling task, which predicts a masked subset of the target image's tokens conditioned on both the multi-modal control signals and the generation target's unmasked tokens. During inference, we adopt Mask-Predict, a NAR generation algorithm [16, 21, 7], which predicts all target tokens at the first iteration and then iteratively re-mask and re-predict a subset of tokens with low confidence scores. To further improve upon the NAR generation algorithm, we exploit the discriminative capability of the BERT architecture [11, 70] and add two estimators (see Figure 2), where one estimator estimates the relevance between the generated image and the control signals, and the other one estimates the image's fidelity. The two estimators help improve the quality of the synthesized image, because at each iteration we can generate multiple samples and keep only the highly-scored one before starting the next iteration. The two estimators also help save

the number of iterations needed, since the algorithm can dynamically terminate if running for more iterations no longer improves the scores.

The extensive experiments on M2C-Fashion, a newly collected clothing dataset with tens of millions of image-text pairs, as well as on Multi-Modal CelebA-HQ [28, 61], a public facial dataset, demonstrate UFC-BERT can synthesize high-quality images that comply with various multi-modal controls.

## 2 Related Works

We have discussed the connection between our work and **Two-Stage Image Synthesis** in the Introduction. In this section, we further discuss related works from other fields.

**Conditional Image Synthesis.** A variety of control signals have been introduced into conditional image synthesis. The class-conditional generation task [1, 39] adopts class labels as control signals. The text-to-image synthesis task [64, 65, 30, 62, 72, 54] further employs natural language descriptions as controls. The image-to-image translation task generates photo-realism images from visual controls, such as a sketch map [17, 60], semantic label map [26, 4, 58], human pose [37] or another image for style transfer [15, 27]. Moreover, image outpainting and inpainting [25, 63] can be regarded as image synthesis conditioned on preservation control signals, where some image blocks of the desired image are already specified and need to be preserved in the generated image. However, these works only utilize one kind of control signal and design their methods customized for each kind of control. Text-guided image manipulation [10, 40, 69, 31, 61] semantically edits an image, where the text description and the original image serve as control signals. But they still fail to unify multiple modalities in a universal form and cannot easily extend to more control modalities. To promote versatility and extensibility, we propose UFC-BERT to unify any number of multi-modal controls.

**Visual-Language Transformer.** With great progress in language tasks [55, 44, 45, 2], the transformer architecture is being rapidly transferred to other fields such as vision [3, 68, 12] and audio [6]. Recently, pretraining visual-language transformer [43, 24, 70, 7, 35, 53, 67] (e.g. multi-modal BERT) has achieved significant improvements on a variety of downstream tasks, e.g. visual question answering, image captioning [70], and text-to-image generation [7]. Among them, the single-stream architecture [52, 32, 46, 5, 43, 24] uses a single transformer to jointly model a pair of text and image, while the two-stream architecture [35, 36, 53] applies two transformers to separately learn the representations of the text and the image, respectively. Our UFC-BERT is also a variant of the single-stream visual-language transformer, but focuses on flexible multi-modal image synthesis instead of multi-modal pretraining.

**Non-Autoregressive Sequence Generation.** Though it is natural to autoregressively predict tokens from left to right when generating a sequence, autoregressive decoding suffers from the slow speed and sequential error accumulation. Thus, the non-autoregressive generation (NAR) paradigm is proposed to avoid these drawbacks in neural machine translation [19, 20, 29, 16], image captioning [14, 22, 70], and speech synthesis [50, 49]. These approaches often employ the bidirectional Transformer (i.e. BERT) as it is not trained with a specific generation order. Our progressive NAR generation algorithm improves upon the Mask-Predict non-autoregressive algorithm [57, 16, 38, 33], by introducing the relevance estimator and the fidelity estimator to facilitate sample selection and dynamic termination.

## 3 UFC-BERT For Multi-Modal Image Synthesis

### 3.1 Background: Two-Stage Image Synthesis

In this section, we review the two-stage architecture [42, 48, 13, 47] for image synthesis.

At the first stage, a codebook $\mathcal{Z} = \{\mathbf{z}_k\}_{k=1}^{K}$ for vector quantization is learned, where $\mathbf{z}_k \in \mathbb{R}^{n_z}$ is the $k$-th code-word in the codebook and $K$ is the number of code-words. An image $\mathbf{X} \in \mathbb{R}^{H \times W \times 3}$ can be transformed into (or from) a collection of code-words $\mathbf{Z} \in \mathbb{R}^{h \times w \times n_z}$. Concretely, a convolutional encoder $E$ first encodes the original image $\mathbf{X}$ as $\hat{\mathbf{Z}} = E(\mathbf{X}) \in \mathbb{R}^{h \times w \times n_z}$. Then an element-wise quantization step $\mathbf{q}(\cdot)$ is applied to each element $\hat{\mathbf{Z}}_{ij}$ to obtain the element's closest code-word $\mathbf{z}_k$, i.e., $\mathbf{q}(\hat{\mathbf{Z}}_{ij}) = \arg\min_{\mathbf{z}_k \in \mathcal{Z}} \|\hat{\mathbf{Z}}_{ij} - \mathbf{z}_k\|$. For reconstruction, a convolutional decoder $D$ is also learned for recovering image $\hat{\mathbf{X}} \in \mathbb{R}^{H \times W \times 3}$ from $\mathbf{Z}$ such that $\hat{\mathbf{X}}$ is close to $\mathbf{X}$. The first stage can be denoted by

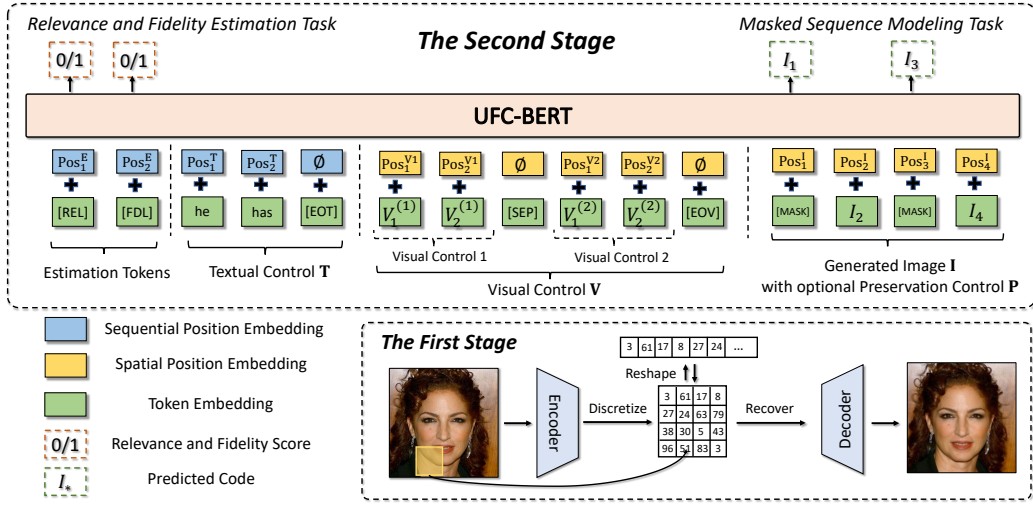

Figure 2: The framework of UFC-BERT, where the textual control (TC), vsiual control (VC), and preservation control (PC), as well as the image to generate, collectively form a sequence of tokens.

$$\mathbf{Z} = \mathbf{q}(E(\mathbf{X})), \ \hat{\mathbf{X}} = D(\mathbf{Z}). \tag{1}$$

Due to the convolutional layers, each of the $h \times w$ elements of $\hat{\mathbf{Z}}$ mainly correlates with an $\frac{H}{h} \times \frac{W}{w}$ block of the image, though its receptive field may be larger if multiple convolutions are stacked.

At the second stage, image $\mathbf{X}$'s quantized representation $\mathbf{Z}$ can be rewritten as a sequence of codes $\mathbf{I} \in \{0, \ldots, |\mathcal{Z}| - 1\}^{N_I}$, composed of $N_I \ (= h \times w)$ indices from the codebook $\mathcal{Z}$. Thus, image synthesis can be formulated as autoregressive sequence generation, i.e. predicting the distribution $\Pr(I_i | \mathbf{I}_{<i}, \mathbf{C})$ of the next token $I_i$ conditioned on the preceding tokens $\mathbf{I}_{<i}$ and the control signals $\mathbf{C}$. The distribution is typically modeled using a unidirectional Transformer. The likelihood is then $\Pr(\mathbf{I}|\mathbf{C}) = \prod_i \Pr(I_i | \mathbf{I}_{<i}, \mathbf{C})$. Parameters are learned by minimizing $\mathcal{L}_{\mathrm{AR}} = \mathbb{E}_{\mathbf{I} \sim data} \left[ - \log \Pr(\mathbf{I}|\mathbf{C}) \right]$.

We focus on improving the second stage. Specifically, the autoregressive paradigm adopted by the existing two-stage works [42, 48, 13, 47] suffers from slow generation speed, fails to capture bidirectional contexts, and cannot fully support preservation control signals. We thus propose UFC-BERT, a novel NAR approach for stage two, to unify any number of multi-modal controls and tackle the shortcomings of AR. As for stage one, we directly follow VQGAN's design [47], which improves upon VQVAE [42] by incorporating a perceptual loss [27] and patch-based adversarial training [26].

## 3.2 Problem Formulation

Conditional image synthesis aims to generate an image that satisfies a set of control signals $\mathbf{C}$. We consider three major modalities of control signals. A *Textual Control (TC)* consists of a sequence of words $\mathbf{T} \in \{0, \ldots, |\mathcal{W}| - 1\}^{N_T}$, where $\mathcal{W}$ is the vocabulary and $N_T$ is the number of words in the text. In the two-stage framework, an image can be converted into a sequence of code-words (i.e. tokens) based on stage one's encoder $E$ and codebook $\mathcal{Z}$. Thus, a *Visual Control (VC)* is denoted by a sequence $\mathbf{V} \in \{0, \ldots, |\mathcal{Z}| - 1\}^{N_V}$ consisting of code-words from the codebook $\mathcal{Z}$, where $N_V$ is the sequence length. Similarly, the target (i.e., the image to generate) is a sequence of code-words $\mathbf{I} \in \{0, \ldots, |\mathcal{Z}| - 1\}^{N_I}$. We support zero, one, or multiple visual controls for flexibility. As for the *Preservation Control (PC)*, it is a sequence of binary masks $\mathbf{P} \in \{0, 1\}^{N_I}$ with the same length as $\mathbf{I}$, where 1 means that the token is known (i.e., $I_i$ is ground-truth if $P_i = 1$) while 0 means the token needs to be predicted. We aim to design a model at the second stage to synthesize the target image's sequence $\mathbf{I}$ conditioned on $\mathbf{C}$, i.e., a combination of any number of control signals from $\{\mathbf{T}, \mathbf{V}, \mathbf{P}\}$.

## 3.3 Model Inputs

As shown in Figure 2, our UFC-BERT modifies the original BERT model [9] to accommodate any number of multi-modal controls. Similar to BERT, the backbone is a multi-layer bidirectional Transformer encoder, enabling the dependency modeling between all input elements. The input sequence of UFC-BERT always starts with two special tokens [REL] and [FDL] for relevance estimation and fidelity estimation, then goes on with the word sequence $\mathbf{T}$ of textual controls, code sequence $\mathbf{V}$ of visual controls, and ends with the code sequence $\mathbf{I}$ of the target image to generate. Two special separation tokens [EOT] and [EOV] are appended to the end of the textual and the visual control sequences, respectively. If there are multiple visual controls, another special token [SEP] is inserted to separate them. The sequence $\mathbf{I}$ of the target image to generate may be partially or fully masked by a special token [MASK]. When the preservation control $\mathbf{P}$ is present and $P_i = 1$, token $I_i$ in $\mathbf{I}$ is always set to the code-word corresponding to the given image block to be preserved.

Each input token's representation is the sum of the position and token embeddings:

**Position Embedding.** Our UFC-BERT learns independent sets of position embeddings for the different kinds of the inputs to achieve better distinguishment between the various modalities. The position embeddings for the word sequence are the same as BERT, i.e., we use sequential position embeddings. For a visual control or the target image, the position embedding of each token is decided according to where this token lies on the $h \times w$ grid, i.e., we use spatial position embeddings.

**Token Embedding.** For textual controls, we use Byte-Pair Encoding [51] to segment each word into sub-words and then learn sub-word embeddings. Each special token, e.g., [REL] or [MASK], is assigned a dedicated embedding. For visual controls and the target image, we learn an embedding for each code-word. We do not directly use the embeddings from stage one's codebook due to the decoupling of the two stages.

## 3.4 Training: Masked Sequence Modeling with Relevance and Fidelity Estimation

As shown in Figure 2, we train UFC-BERT via masked sequence modeling, i.e., predicting the masked tokens in the target image conditioned on the controls. A relevance estimator and a fidelity estimator are also trained in the process, and will be key to our progressive NAR generation algorithm.

**Task 1: Masked Sequence Modeling.** This task is similar to Masked Language Modeling (MLM) in BERT, but incorporates multi-modal control signals when predicting the masked tokens. To construct training samples, we mask parts of the target image $\mathbf{I}$ to predict using four strategies: (1) randomly decide the number of tokens to mask, and then randomly mask the desired number of tokens; (2) mask all tokens; (3) mask the tokens within some boxed areas of the image, where the number of boxes and the box sizes are randomly decided; (4) mask the tokens outside some random boxed areas of the image. We use the four strategies with probability 0.70, 0.10, 0.10, and 0.10, respectively. To construct multi-modal control signal $\mathbf{C}$ for each training sample, there are four different combinations: *<TC, VC>, <TC>, <VC>, <empty>*, where *<TC, VC>* means the textual and visual controls are simultaneously employed, *<TC>* or *<VC>* means only a textual or visual signal is used, and *<empty>* means no textual or visual control is present. Note that the preservation control is already included in the masked sequence modeling task. Since our dataset does not contain ground-truth pairs of visual controls and target images, we crop one or multiple regions of a target image to construct VC for the target image. Because image synthesis from solely textual controls is more challenging than from other signals, we use the four combinations with probability 0.20, 0.55, 0.20, 0.05, respectively, where textual controls get more attention. We feed UFC-BERT's outputs at each position of $\mathbf{I}$ into a softmax classifier over the codebook $\mathcal{Z}$, which produces a probability score $Y_i = \Pr(I_i | \mathbf{I}_U, \mathbf{C})$ for each position $i \in M$, where $M$ is the set of masked positions and $U$ is the unmasked set. Finally, the masked sequence modeling task minimizes the softmax cross-entropy loss $\mathcal{L}_{\text{MSM}} = \mathbb{E}_{\mathbf{I}_M, \mathbf{I}_U} \left[ -\log \Pr(\mathbf{I}_M | \mathbf{I}_U, \mathbf{C}) \right]$, where $\Pr(\mathbf{I}_M | \mathbf{I}_U, \mathbf{C}) = \prod_{i \in M} Y_i$.

**Task 2: Relevance Estimation.** This task is to learn a binary classifier that judges whether the generated image is relevant or irrelevant to the given multi-modal control $\mathbf{C}$. Briefly, we add a linear layer on the output corresponding to the special token [REL]. The linear layer outputs a scalar representing the logit, and a binary cross-entropy loss $\mathcal{L}_{\text{REL}}$ is added. During training, the training samples from Task 1 serve as the positive instances (i.e. relevant pairs). We construct negative instances (i.e. irrelevant pairs) by swapping the control signals of two training samples.

**Task 3: Fidelity Estimation.** This task aims to distinguish whether the generated image is realistic from the view of human visual cognition. Similar to relevance estimation, we feed the output corresponding to [FDL] into a linear layer for binary classification and add another binary cross-entropy loss $\mathcal{L}_{\text{FDL}}$. Since the low-fidelity images (i.e. negative instances) do not exist in the dataset, we run UFC-BERT from previous epochs to synthesize images based solely on textual control signals, and use the synthesized images as negative instances.

We combine the three tasks' losses to train UFC-BERT, i.e.,

$$\mathcal{L}_{\text{UFC-BERT}} = \lambda_1 \mathcal{L}_{\text{MSM}} + \lambda_2 \mathcal{L}_{\text{REL}} + \lambda_3 \mathcal{L}_{\text{FDL}}, \tag{2}$$

where $\lambda_1$, $\lambda_2$ and $\lambda_3$ are set to 1.0, 0.5, and 0.5 to balance the three losses. The masked sequence modeling task ignores the negative instances from the other two tasks, i.e., irrelevant pairs or unrealistic instances. And the fidelity estimation task is added only after a certain number of epochs.

### 3.5 Inference: Progressive Non-Autoregressive Generation

We design a Progressive Non-Autoregressive Generation (PNAG) algorithm for conditional image synthesis after training, which improves upon Mask-Predict [16, 21, 7]. Mask-Predict predicts all target tokens when given a fully-masked sequence at the first iteration, and then iteratively re-mask and re-predict a subset of tokens with low probability scores for a constant number of iterations. However, Mask-Predict cannot ensure the efficacy of multi-modal controls and the fidelity of the synthesized images, and requires determining the number of iterations. Our PNAG tackles its drawbacks via sample selection and dynamic termination, based on the relevance and fidelity estimators.

Each iteration of our PNAG algorithm consists of a *Mask* step and then a *Predict* step. Let $\mathbf{I}^{(t,in)} = (I_1^{(t,in)}, \ldots, I_{N_I}^{(t,in)})$ and $\mathbf{I}^{(t,out)} = (I_1^{(t,out)}, \ldots, I_{N_I}^{(t,out)})$ be the state of the target image's sequence before and after the $t$-th iteration, respectively. The tokens in $\mathbf{I}^{(0,out)}$ for $t = 0$ is all set to [MASK] except for the positions that are controlled by the preservation signals, i.e., except for $I_i^{(0,out)}$ that has $P_i = 1$. If a preservation control is present, i.e. $P_i = 1$, we always set $I_i^{(t,in)}$ and $I_i^{(t,out)}$ for all $t$ to be the code-word that corresponds to the provided image block to be preserved.

**Mask Step.** At the beginning of iteration $t$ ($t \geq 1$), we construct the input sequence $\mathbf{I}^{(t,in)}$ by (re-)masking a subset of tokens in the generated sequence $\mathbf{I}^{(t-1,out)}$ from the last iteration. Similar to beam search, we construct $B$ parallel input sequences $\{\mathbf{I}_1^{(t,in)}, \ldots, \mathbf{I}_B^{(t,in)}\}$ at each iteration. Specifically, we re-mask $n$ tokens of $\mathbf{I}^{(t-1,out)}$ to produce each $\mathbf{I}_b^{(t,in)}$. We first sample $N_I - n$ tokens from a multinomial distribution $\Pr^{(t,in)}$ proportional to the probability scores $\mathbf{Y}^{(t-1)} = \{Y_i^{(t-1)}\}_{i=1}^{N_I}$ (see Equation 3), computed by $\Pr^{(t,in)} = \text{Softmax}(\mathbf{Y}^{(t-1)})$. And other tokens are re-masked and re-predicted at the next Predict Step. Here $n = N_I \cdot (\beta + \frac{T-t}{T-1} \cdot (\alpha - \beta))$, where $\alpha$ is the initial mask ratio, $\beta$ is the minimum mask ratio, and $T$ is the maximum possible number of iterations, such that the number of tokens to re-mask gradually decreases after every iteration.

**Predict Step.** Given the control $\mathbf{C}$ and an input sequence $\mathbf{I}_b^{(t,in)}$, UFC-BERT estimates a distribution $\Pr(\hat{I}_i | \mathbf{I}_b^{(t,in)}, \mathbf{C})$ for each masked position $i$. UFC-BERT also estimates the relevance score $S_b^R$ and fidelity score $S_b^F$ regarding the image that it is about to synthesize, and summarizes the scores into a comprehensive score $S_b^{(t)} = \sigma S_b^R + (1 - \sigma) S_b^F$, where $\sigma$ is a coefficient for adjusting the importance of the two. We perform *sample selection* based on $S_b^{(t)}$, i.e., we select the $b$-th sequence $\mathbf{I}_b^{(t,in)}$ with the highest $S_b^{(t)}$, discard the others, and then generate $\mathbf{I}^{(t,out)}$ based on the selected $\mathbf{I}_b^{(t,in)}$ as follows:

$$I_i^{(t,out)} \sim \Pr(\hat{I}_i | \mathbf{I}_b^{(t,in)}, \mathbf{C}), \qquad Y_i^{(t)} \leftarrow \Pr(\hat{I}_i = I_i^{(t,out)} | \mathbf{I}_b^{(t,in)}, \mathbf{C}), \tag{3}$$

where each token $I_i^{(t,out)}$ is sampled from the multinomial distribution $\Pr(\hat{I}_i | \mathbf{I}_b^{(t,in)}, \mathbf{C})$ and the corresponding probability is assigned to $Y_i^{(t)}$. Note that we predict tokens for *all* masked positions regardless of the predictions' confidence. We also implement *dynamic termination* based on $S_b^{(t)}$. Specifically, if the current iteration's score $S_b^{(t)}$ is higher than $S_{max}$ (initialized as zero), we set $S_{max}$ to $S_b^{(t)}$ and record the current iteration as $t_{max}$. If $S_{max}$ does not increase after three consecutive iterations, we select $\mathbf{I}^{(t_{max},out)}$ as the final result and terminate our generation algorithm.

| M2C-Fashion | | | | Multi-Modal CelebA-HQ | | | |
|---|---|---|---|---|---|---|---|
| TC | VC | PC | image | TC | VC | PC | image |
| 真丝纱网拼接A字连衣裙 Silk gauze stitching A-line dress | ∅ | ∅ | | She has high cheekbones, and blond hair. She is attractive, and young. | ∅ | ∅ | |
| ∅ | | ∅ | | ∅ | | ∅ | |
| ∅ | ∅ | | | ∅ | ∅ | | |
| 灰色针织短袖T恤 Grey knitted short sleeve T-shirt | | ∅ | | He has the receding hairline and wavy hair. | | ∅ | |
| 夏季轻薄真丝连衣裙两件套装 Two-piece summer light silk dress | ∅ | | | She wears heavy makeup, and lipstick. She has arched eyebrows, big lips, and pointy nose. | ∅ | | |
| ∅ | | | | ∅ | | | |
| 翻领垫肩橙色带口袋西装大衣 Lapel shoulder pad orange suit coat with pockets | | | | The person has mustache, bushy eyebrows, narrow eyes, and high cheekbones. | | | |

Figure 3: Images generated by our UFC-BERT under various combinations of textual controls (TC), visual controls (VC), and preservation controls (PC). Please see the supplemental material for more showcases, where we also include a study on the diversity of the images generated by UFC-BERT and analyze how the multiple control signals interfere with each other.

## 4 Experiments

### 4.1 Datasets and Hyperparameters

In experiments, we focus on two practical fields of image synthesis: fashionable clothing and human faces. We collect a very large-scale clothing dataset M2C-Fashion with Chinese text descriptions, which contains tens of millions of image-text pairs, much larger than the commonly used text-to-image datasets COCO [34] and CUB [56]. Details of the dataset are provided in the supplementary material. We additionally use another high-resolution facial dataset Multi-Modal CelebA-HQ [28, 61].

Following the model setting of VQGAN [13], we use the $256 \times 256$ image size on the two datasets and transform each image to a discrete sequence of $16 \times 16$ codes, where the codebook size $|\mathcal{Z}|$ is set to 1024. For the BERT model, we set the number of layers, hidden size, and the number of attention heads to 24, 1024, and 16, respectively. Our UFC-BERT has 307M parameters, same as the Transformer used by VQGAN. As for hyper-parameters of PNAG, we set the parallel decoding number $B$ to 5 and the balance coefficient $\sigma$ to 0.5. We set the initial mask ratio $\alpha$, the minimum mask ratio $\beta$, and the maximum iteration number $T$ to 0.8, 0.2, and 10, respectively.

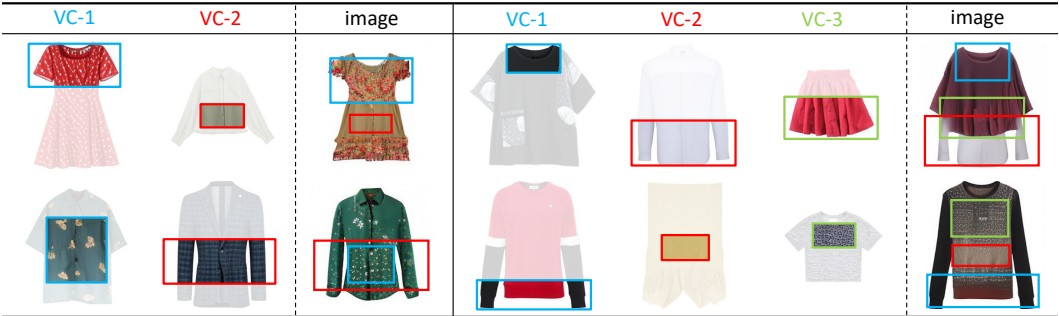

Figure 4: Image synthesis with multiple visual controls, where we crop regions from $2 \sim 3$ images to serve as the visual controls. UFC-BERT synthesizes images that naturally fuse the visual elements.

Table 1: Comparisons with GAN baselines for text-to-image synthesis on Multi-Modal CelebA-HQ.

| Method | AttnGAN [62] | ControlGAN [30] | DF-GAN [54] | DM-GAN [71] | TediGAN [61] | UFC-BERT (our) |
|---|---|---|---|---|---|---|
| FID ↓ | 125.98 | 116.32 | 137.60 | 131.05 | 106.37 | **66.72** |
| LPIPS ↓ | 0.512 | 0.522 | 0.581 | 0.544 | 0.456 | **0.448** |

Table 2: Comparisons with the autoregressive two-stage method VQGAN for text-to-image synthesis. ↓ means the lower the better, while ↑ means the opposite. We evaluate speed on the same V100 GPU.

| Datasets | Methods | Automatic Metrics | | | | Human Pairwise Study | | Inference Speed |
|---|---|---|---|---|---|---|---|---|
| | | FID↓ | LPIPS ↓ | PSNR↑ | SSIM↑ | Relevance | Fidelity | |
| M2C-Fashion | VQGAN (AR) | 12.48 | 0.483 | 10.80 | 0.56 | 38.6% | 44.2% | 8.73 sec/sample |
| | UFC-BERT (NAR) | **11.53** | **0.461** | **13.14** | **0.58** | **61.4%** | **55.8%** | **0.81** sec/sample |
| Multi-Modal CelebA-HQ | VQGAN (AR) | **52.63** | 0.503 | 8.98 | 0.28 | 42.7% | 46.9% | 8.66 sec/sample |
| | UFC-BERT (NAR) | 66.72 | **0.448** | **9.56** | **0.29** | **57.3%** | **53.1%** | **0.79** sec/sample |

## 4.2  Flexibility of Multi-Modal Controls for Conditional Image Synthesis

In this section, we qualitatively verify the synthesis ability of UFC-BERT with three modalities of control signals, i.e., textual, visual, and preservation controls. The textual controls are the texts paired with the images, which are already provided by the two datasets, while the visual controls are code sequences of cropped regions, e.g. regions that represent logos or texture of clothes.

In Figure 3, we synthesize images conditioned on combinations of the three types of control signals. The results demonstrate UFC-BERT can unify any number of multi-modal controls to synthesize high-quality images. Further, UFC-BERT supports one or multiple visual controls for more flexible synthesis, as shown in Figure 4 where we generate images given 2~3 visual controls. We observe that UFC-BERT can reasonably fuse multiple visual elements and produce a harmonious image.

## 4.3  Quantitative Comparison to Existing Methods for Text-to-Image Synthesis

In this section, we investigate how our UFC-BERT quantitatively compares to existing models. Considering most existing methods only utilize one control signal, we select the most common and challenging task *text-to-image synthesis* to compare the synthesis ability.

First, we compare our UFC-BERT with GAN-based text-to-image models AttnGAN [62], Control-GAN [30], DF-GAN [54], DM-GAN [71] and TediGAN [61] on the Multi-Modal CelebA-HQ dataset. For evalution, we adopt two automatic metrics FID [23] and LPIPS [66]. We report the results on Table 1 and our UFC-BERT achieves the best performance on the two metrics, even outperforming the TediGAN that uses slow and complex instance-level optimization. This demonstrates the two-stage architecture and non-autoregressive generation of UFC-BERT are suitable for text-to-image synthesis.

Women's striped long-sleeved orange shirt. 女式条纹长袖橙色衬衫

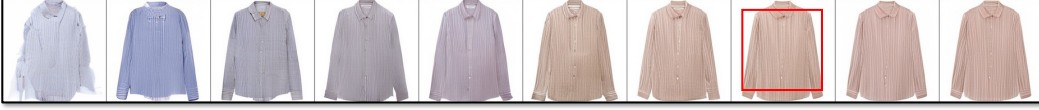

This woman has wavy hair and is wearing earrings, and lipstick.

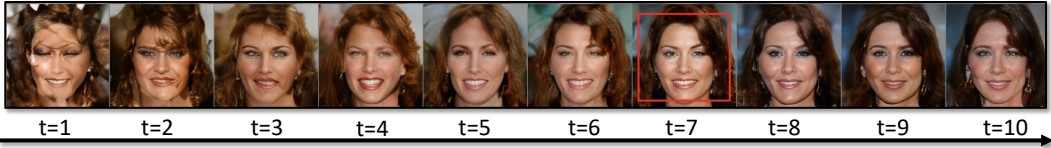

t=1 t=2 t=3 t=4 t=5 t=6 t=7 t=8 t=9 t=10

Figure 6: The iterative inference process of our PNAG algorithm. The red bounding box means the image has the highest comprehensive score and is selected as the final output result.

Besides, we compare our UFC-BERT with the autoregressive two-stage method VQGAN from three aspects: (i) the automatic metrics FID for image quality, as well as LPIPS, PSNR [59] and SSIM [59] for the similarity between the generated image and the ground truth; (ii) the Relevance and Fidelity metrics are evaluated through a user study, where the users are asked to judge which model's output is more relevant to the textual descriptions, and more photorealistic; (iii) the synthesis speed of the two approaches. Note that the autoregressive inference implementation of VQGAN has been optimized by caching the preceding computation as in Transformer-XL [8], and UFC-BERT and VQGAN have the same parameter number (307M) for fair comparison.

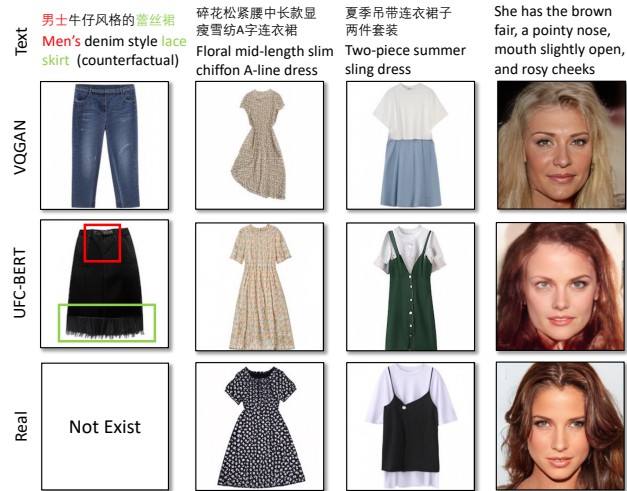

Figure 5: Typical examples of UFC-BERT and VQGAN for text-to-image synthesis, including a counterfactual case.

For the user study, the two models receive the same textual signals, and each generates 50 images. We collect the pairwise comparison results from five volunteers.

As shown in Table 2, our UFC-BERT achieves better performance for almost all criteria with about $11\times$ speedup. This suggests our non-autoregressive UFC-BERT with progressive NAR generation algorithm can synthesize high-fidelity images relevant to textual descriptions. As for the FID metric, UFC-BERT outperforms VQGAN on M2C-Fashion, but has worse performance on Multi-Modal CelebA-HQ, it may be due to the fact that the autoregressive VQGAN can more easily memorize the pattern of a small dataset (only 30,000 facial images). In Figure 5, we further show typical generated examples to intuitively display the difference between the two approaches, including a case of counterfactual generation. We find that UFC-BERT can synthesize high-quality images, even for the counterfactual case.

## 4.4 The Effectiveness of Our Progressive NAR Generation Algorithm

In this section, we first visualize in Figure 6 the iterative process of our PNAG inference method based on the relevance and fidelity estimators. The images with red bounding boxes are the final outputs that match the textual control signals. We can find that the fidelity and relevance of the images increase after a few iterations, verifying our PNAG algorithm can guide the inference process towards a better direction and synthesize more realistic images that match the control signals.

Table 3: Ablation studies of our PNAG inference algorithm. PNAG(w/o. REF) and PNAG(w/o. FDL) set $B$ to the default value 5. MNAG is the original Mask-Predict algorithm [16].

| Dataset | Metrics | MNAG [16] | PNAG(w/o. REF) | PNAG(w/o. FDL) | PNAG($B$=1) | PNAG($B$=5) | PNAG($B$=10) |
|---------|---------|-----------|----------------|----------------|-------------|-------------|--------------|
| M2C-Fashion | FID ↓ | 14.77 | 12.17 | 13.14 | 12.72 | 11.53 | **11.14** |
| | LPIPS ↓ | 0.488 | 0.477 | 0.469 | 0.479 | 0.461 | **0.456** |
| Multi-Modal CelebA-HQ | FID ↓ | 72.04 | 68.90 | 70.32 | 69.49 | 66.72 | **65.30** |
| | LPIPS ↓ | 0.514 | 0.469 | 0.463 | 0.475 | 0.448 | **0.445** |

We then conduct ablation studies of PNAG. As shown in Table 3, we develop three ablated inference methods PNAG(w/o. REF), PNAG(w/o. FDL) and MNAG, where PNAG(w/o. REF) and PNAG(w/o. FDL) discard the relevance estimator and the fidelity estimator, respectively, and MNAG is the original Mask-Predict method [16] without any estimator. The results demonstrate that the two estimators effectively utilize the discriminative capability of UFC-BERT and do help improve the synthesis quality. Additionally, we vary the crucial hyper-parameter of PNAG $B$ (i.e. the parallel decoding number during inference) from 1 to 10, and the results in Table 3 show that a larger $B$ is beneficial to the synthesis quality.

## 5 Conclusions

We proposed UFC-BERT to unify any number of multi-modal controls in a universal form for conditional image synthesis. We utilized non-autoregressive generation to improve inference speed, enhance holistic consistency, and support preservation controls. Further, we designed a progressive generation algorithm based on relevance and fidelity estimators to ensure relevance and fidelity.

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
