# UFC-BERT: Unifying Multi-Modal Controls for Conditional Image Synthesis

**Zhu Zhang[†], Jianxin Ma[†], Chang Zhou[†], Rui Men[†], Zhikang Li[†],**
**Ming Ding[‡], Jie Tang[‡], Jingren Zhou[†], and Hongxia Yang[†]**
[†]DAMO Academy, Alibaba Group, [‡]Tsinghua University
{zhangzhu950310}@gmail.com
{jason.mjx, ericzhou.zc, yang.yhx}@alibaba-inc.com

## A    Details of the M2C-Fashion Dataset

We collect the large-scale M2C-Fashion dataset from the largest Chinese shopping website Taobao. First, we obtain the display images and captions of products under the clothing category. We then only retain the white background images and filter out those images with complex backgrounds. This operation avoids the unnecessary complexity for the various backgrounds during image synthesis and allows the model to focus on the details of the clothes. Next, we filter out the images with captions less than five words to guarantee the textual descriptions have abundant information. After it, we remove duplicate images and the images which resolution is less than 64*64. Finally, we convert the resolution of all images to 256*256 for experiments. There are 10,855,753 image-caption pairs in total, where 10,845,753 pairs are used for training, 5,000 pairs for validation, and 5,000 pairs for testing.

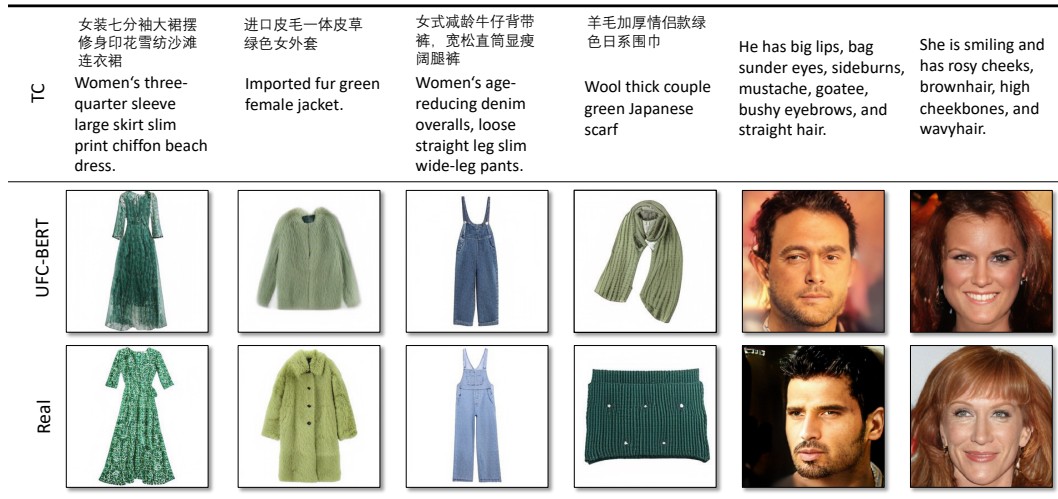

Figure 1: Image synthesis with textual controls.

## B    More Examples for Conditional Image Synthesis

In this section, we display more images generated by our UFC-BERT under various combinations of control signals. Instead of deliberately selecting high-quality images, we randomly select some

35th Conference on Neural Information Processing Systems (NeurIPS 2021).

examples to show the real synthesis ability. Here we apply the three most commonly-used control combinations. Figure 1 shows the generated images only with textual controls, Figure 2 gives the synthesized images under the combinations of textual and visual controls, and Figure 3 displays the images with textual and preservation controls.

## C   Diversity of Conditional Image Synthesis

In this section, we show the diverse images generated from the same control signals. To produce the multiple results, we only need to retain multiple sequences at the first **Predict Step** during progressive inference and then conduct parallel synthesis. Concretely, we generate 5 diverse images for given controls, where we select textual controls and the combinations of textual and visual controls. We show the results in Figure 4. We can observe each generated image complies with multi-modal controls but has obvious differences compared to other images under the same controls.

## D   Interference of Multiple Control Signals

In this section, we investigate the generated images when different control signals have semantic conflicts. We select the textual controls and visual controls, and observe the interference between them on the clothing color and style. We show the results in Figure 5. We observe that when the semantic interference occurs in colors, the color of synthesized images may only be based on a control signal, or there may be a mixture of multiple colors. If the semantic conflict occurs in the style, the generated images often follow the semantics of textual controls, as the visual control is a visual element rather than a complete image.

## E   Broader Impact

This paper introduces a new architecture UFC-BERT to unify any number of multi-modal controls for flexible conditional image synthesis. As shown in experiments, UFC-BERT can be applied to the clothing design. Thus, this research can promote the development of smart manufacturing, facilitate personalized clothing customization, and assist clothing designers to improve efficiency. And the face synthesis of UFC-BERT can be applied to game character design for avoiding infringement of personal privacy. The synthesized face images can also be used as augmented data for improving the performance of face recognition models. However, the powerful capability for clothing synthesis may be used to plagiarize trendy clothing. The synthesized face may also be used to deceive the face detection system and bring some negative societal effects.


| TC | VC | Image | TC | VC | Image |
|---|---|---|---|---|---|
| 夏季仙女系少女蓝色短袖连衣裙

Summer fairy girl blue short-sleeved dress. | | | 夏季宽松带帽短袖学生T恤

Summer loose hooded short-sleeved student T-shirt. | | |
| 黑色复古优雅花鸟刺绣中长款网纱连衣裙

Black retro elegant flower and bird embroidery mid-length mesh dress. | | | The person has brown hair and pointy nose. She wears heavy makeup. | | |

Figure 2: Image synthesis with textual and visual controls.

| TC | PC | Image | TC | PC | Image |
|---|---|---|---|---|---|
| 宽松字母印花圆领套头女卫衣

Loose letter print round neck pullover women's sweater. | | | The person wears heavy makeup. She has mouth slightly open, big lips, and high cheekbones. | | |
| 12岁小女孩白色长袖衬衣

12 year old girl in white long-sleeved shirt. | | | This young person has brown hair, oval face, and mouth slightly open. | | |

Figure 3: Image synthesis with textual and preservation controls.

| TC | VC | Diverse Images |
|---|---|---|
| 女性宽松长袖带口袋西装上衣

Women's loose long-sleeved suit jacket with pockets. | ∅ | |
| The woman has big nose, big lips, and high cheekbones and wears heavy makeup. She is smiling. | ∅ | |
| 仿水貂绒开衫，长袖针织毛衣外套

Imitation mink fleece cardigan, long-sleeved knitted sweater coat. | | |
| 时尚休闲宽松圆领长袖羊毛衫

Fashion casual loose long-sleeved round neck sweater. | | |

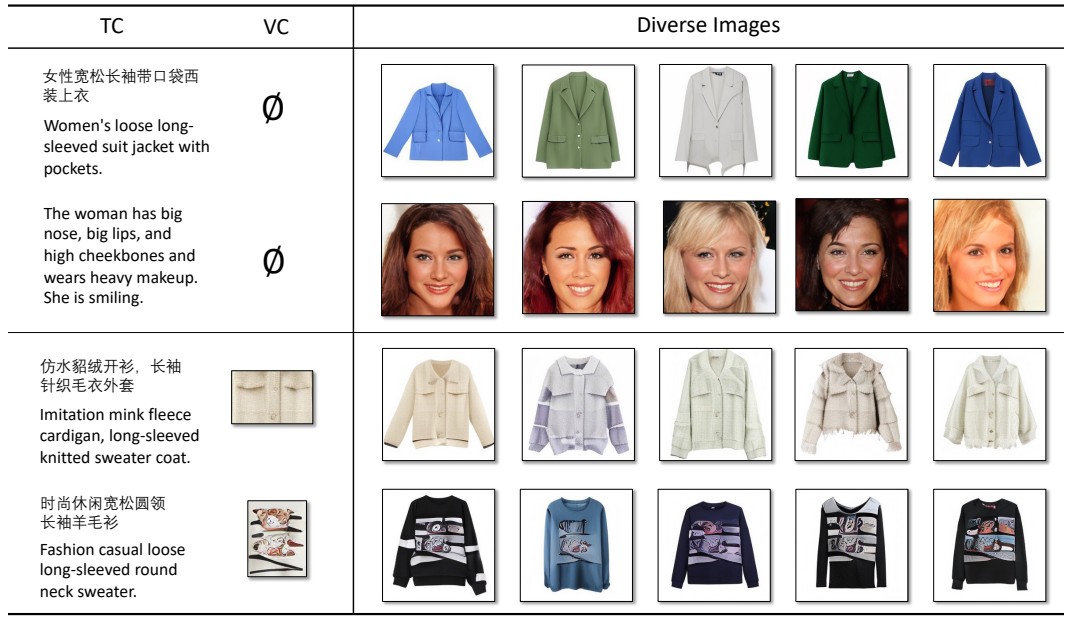

Figure 4: Diverse image synthesis under the same controls.

| TC | VC | Image | TC | VC | Image |
|---|---|---|---|---|---|
| 红色圆领短袖衬衫

Red round neck short sleeve shirt. | | | 红色圆领短袖衬衫

Red round neck short sleeve shirt. | | |
| 蓝色圆领长袖羊毛衫

Blue round neck long sleeved sweater. | | | 蓝色圆领长袖羊毛衫

Blue round neck long sleeved sweater. | | |

Figure 5: Interference between textual controls and visual controls.