# OpenReview forum: "UFC-BERT: Unifying Multi-Modal Controls for Conditional Image Synthesis"
_NeurIPS.cc/2021/Conference — NeurIPS 2021 Poster_

### Official Review · Reviewer_BADA · 2021-07-06

**Rating:** 7
**Confidence:** 4

**Summary:**

### Summary
The paper works on conditional image synthesis using a transformer-based network. The proposed method uses one network to combine multi-modal conditional signals, such as textual information and visual information, for image generation. The proposed non-autoregressive training strategy is validated on two datasets, the collected clothing dataset and Multi-Modal CelebA-HQ, and shows better results than GAN-based methods and a transformer-based method.

### Strengths
- The proposed method can utilize multi-modal controls for image synthesis and show convincing synthesized images containing input condition information. The image generation results conditioned on multi-modal inputs are interesting and inspiring.
- While recent autoregressive-based methods demonstrate good text-to-image generation results using transformers, the paper shows that non-autoregressive transformers (e.g., BERT used in the paper) can also be trained for generating a sequence of image tokens.
- The careful design of the relevance and fidelity estimation tasks is novel and helps the training of the network.


**Limitations And Societal Impact:**

It would be better if authors could discuss more about the limitation of the methods.

**Main Review:**

I have some questions regarding the methods and experiment details.

### Methods
- On Ln. 189, the paper mentions that there could no textual or visual control. In this case, how could the network be trained if all the image tokens are masked (case (2) on Ln. 183)?
- For constructing negative pairs in Task 2 (Relevance Estimation), do authors use images without masked tokens, or could the tokens still be masked? If the tokens still could be masked, do authors perform Task 1 for the generated missing tokens?
- I'm a little confused about Ln. 232-233. The method seems to mask tokens with high probability scores by sampling from a multinomial distribution. But my understanding is that the method aims to mask tokens with low probability scores. Is the description on Ln. 232-233 accurate?

### Experiments
- For experiments on M2C-Fashion and Multi-Modal CelebA-HD, do authors train the network from scratch or from pre-trained models? If from pre-trained models, how much performance improvement compared with training from scratch?
- I wonder whether authors have tried any experiments on the CUB200 bird dataset.
- For training the VQGAN transformer, could the authors provide more details on how to process the textual information? Is it the same for training UFC-BERT?
- On Ln.309-310, the authors mention autoregressive VQGAN can more easily remember the small dataset. I wonder would authors explain why autoregressive training can more easily remember the dataset?

### Minor
Will the code and dataset be released?

===============

Thanks authors for submitting the rebuttal. The rebuttal addresses most of my concerns and answers the questions from other reviewers. Although I am concerned about the performance of the proposed framework on different datasets (the paper only shows two datasets), the paper introduces a new way for generating images using multi-modalities, and it is more flexible than auto-regressive transformers. I believe the method can contribute to more potential applications.

**Time Spent Reviewing:**

5

---

> ### Author Response · Authors · 2021-08-10
> **Author Response to Reviewer BADA**
>
>
> We’d like to thank the reviewers for valuable comments.
>
> _____
>
> ### **Methods**
>
> **Q1. On Ln. 189, the paper mentions that there could no textual or visual control. In this case, how could the network be trained if all the image tokens are masked (case (2) on Ln. 183)?**
>
> In fact, we set a rule to avoid adopting case (2) on Ln. 183 if there are no textual and visual controls. Thanks for your comment and we will supplement the detail in revision.
>
> _____
>
> **Q2. For constructing negative pairs in Task 2 (Relevance Estimation), do authors use images without masked tokens, or could the tokens still be masked? If the tokens still could be masked, do authors perform Task 1 for the generated missing tokens?**
>
> For negative pairs in Task 2, the tokens are still masked. This is because during inference, the relevance estimator needs to give the relevance score based on the masked sequence, thus we keep the input form consistent during training and inference. Likewise, we mask the token sequence of the target image for negative instances in Task 3. But as illustrated on Ln.212-214, Task 1 (masked sequence modeling task) ignores the negative instances from Task 2 and 3, i.e. we do not perform Task 1 for irrelevant pairs or unrealistic images.
>
> _____
>
> **Q3. I'm a little confused about Ln. 232-233. The method seems to mask tokens with high probability scores by sampling from a multinomial distribution. But my understanding is that the method aims to mask tokens with low probability scores. Is the description on Ln. 232-233 accurate?**
>
> Thanks for pointing out the unclear descriptions of sampling the re-masked tokens in Line232-Line233. Our Mask Step iteratively re-masks a subset of tokens with low probability scores. We modify Line232-Line234 as follows:
>
>
>
> > Specifically, we re-mask $n$ tokens of ${\bf I}^{(t-1, out)}$ to produce each ${\bf I}^{(t, in)}_{b}$. We first sample $N_I-n$ tokens from a multinomial distribution ${Pr}^{(t,in)}$ proportional to the probability scores ${\bf Y}^{(t-1)} = ({Y}^{(t-1)}_1, {Y}^{(t-1)}_2, \dots, {Y}^{(t-1)}_N)$, computed by ${Pr}^{(t,in)} = Softmax({\bf Y}^{(t-1)})$.
> And other tokens are re-masked and re-predicted at the next Predict Step. Here $n = N_I \cdot (\beta + \frac{T-t}{T-1}\cdot({\alpha}-\beta))$, where $\alpha$ is the initial mask ratio, $\beta$ is the minimum mask ratio, and $T$ is the maximum possible number of iterations, such that the number of tokens to re-mask gradually decreases after every iteration.
>
> Note that the formula of the re-masked number $n$ is also corrected.
>
> _____
>
> ### **Experiments**
>
>
> **Q4. For experiments on M2C-Fashion and Multi-Modal CelebA-HD, do authors train the network from scratch or from pre-trained models? If from pre-trained models, how much performance improvement compared with training from scratch?**
>
> We train the network from scratch. Actually, we can utilize a pre-trained model to initialize our UFC-BERT, where the pre-trained model is learnt on large-scale open-domain data. We think a large-scale pre-trained model can accelerate the convergence of UFC-BERT and can also improve the performance of UFC-BERT on small-scale datasets.
>
> _____
>
> **Q5. I wonder whether authors have tried any experiments on the CUB200 bird dataset.**
>
> During the rebuttal period, we have applied the UFC-BERT to the CUB200 bird dataset. At present, UFC-BERT achieve the 4.71$\pm$0.06 Inception Score and 15.43 FID on CUB200. The results are slightly lower than the current SOTA model. This is because Transformer-based model requires large-scale data to achieve better performance, but the CUB200 dataset only includes 8,855 images for training. Moreover, the current results of UFC-BERT may be further improved if there is sufficient time to adjust the model.
>
> _____
>
> **Q6. For training the VQGAN transformer, could the authors provide more details on how to process the textual information? Is it the same for training UFC-BERT?**
>
> Yes, for training the VQGAN transformer, we process the textual information same as training UFC-BERT. We will add the details in revision.
>
> _____
>
> **Q7. On Ln.309-310, the authors mention autoregressive VQGAN can more easily remember the small dataset. I wonder would authors explain why autoregressive training can more easily remember the dataset?**
>
> Because the autoregressive training/inference will generate the t-th token based on the input of the (t-1)-th token. The pattern from the (t-1)-th token to the t-th token is easier to capture. On the contrary, the non-autoregressive training/inference in UFC-BERT requires to simultaneously predict all masked tokens and does not depend on the last token. Thus, non-autoregressive generation needs more data to capture the complicated patterns. We will add more discuss in revision.
>
> _____
>
> ### **Minor**
>
> **Q8. Will the code and dataset be released?**
>
> Of course, we will release the large-scale dataset M2C-Fashion and our code if accepted.

---

### Official Review · Reviewer_wtTT · 2021-07-14

**Rating:** 7
**Confidence:** 4

**Summary:**

This paper proposes a two-stage non-autoregressive (NAR) transformer-based approach for multi-modal conditional image generation in the discrete latent space of a pretrained VQGAN, called UFC-BERT. More specifically, the proposed model draws inspiration from BERT [1] and Mask-Predict [2] and uses masked language (token) modeling to (i) incorporate bidirectional context and (ii) significantly speed up generation over autoregressive baselines. Sampling from the masked model is performed via an extension of the progressive mask-predict algorithm, where two newly introduced discriminative heads of the transformer model guide the sampling process to generate samples that are consistent with the given conditioning and have high fidelity.

**Limitations And Societal Impact:**

Societal implications are sufficiently discussed. However, I find it interesting that regarding the facial manipulation possibilities of the proposed approach, the only "negative" point raised is the possible deception of facial recognition systems. Depending on one's perspective and culture, one could probably assume the complete opposite here.

There is no section specifically addressing the limitations of the proposed approach.


**Main Review:**

__Strengths:__

The paper presents high-quality image synthesis results with a BERT-style model, previously unseen with these type of models.
Furthermore, the paper does a great job at motivating the proposed solution (especially in line 43-56).
The proposed architecture and sampling process is simple but effective. Additionally, the two discriminative tasks (relevance and fidelity estimation) can be effectively trained via swapping of the conditioning information (relevance) and replay of samples from previous model checkpoints (fidelity), which is a nice feature. Using the BERT-formulation allows to condition on arbitrary patches of images (in contrast to the "completion" property of autoregressive models), where especially Fig. 4 provides impressive results on mixing different patches into a coherent image.

The model is described in detail in Sec. 3 and it should be easy to re-implement, given the provided information.

__Weaknesses:__

- What's the purpose of Figure 1? Neither sketch-to-image nor artistic style transfer are demonstrated with UFC-BERT. This should be clarified in the corresponding caption.
- It remains somewhat unclear how the cropping for VC and PC works. Is this performed in the latent space of the VQGAN, and correspondences are assigned via the mentioned $\frac{H}{h} \times \frac{W}{w}$ token-pixel correlations? If so, what is the effect of the non-local block (=attention layer) present in the VQGAN? Furthermore, crops can then only be made with a stride of 16 pixels.
- The paper repeatedly states that the relevance and fidelity estimators are "key" to the generation algorithm, but does not provide experiments to justify this claim.
- No metric that explicitly measures correspondence between generated images and corresponding textual descriptions is provided, the paper rather takes a detour by measuring similarity of ground-truth images and samples. However, since there is not a one-to-one correspondence between textual description and image, these metrics (LPIPS, SSIM, PSNR) might not be a good fit. On the contrary, LPIPS has been used to measure _diversity_ of generated samples, where higher scores are better [3]. Using R-precision as in [4] or CLIP-scores [5] could be a simple fix.

__Originality:__
The proposed method of sampling latent tokens learned via vector-quantized representation learning (VQVAE/VQGAN) with a BERT-style model is new and allows for applications such as image completion or harmonization of different patches. Furthermore, multi-modal control has been underexplored in the context of VQVAE/VQGAN models.

__Clarity/Quality:__
The presentation is clear, and, as described above (strengths), the model should be easy to re-implement.

__Significance:__
Given the promising results on face and fashion datasets, the approach could have potential to serve as a generic neural image manipulation/creation mechanism when trained in a large-scale setting.

__Mixed Comments/Questions/Suggestions:__
- How does the diversity of samples generated with IFC-BERT compare to an explicit AR-likelihood model?
- I would rather call the "counterfactual case" (l. 312 and Fig. 5) a "conflicting prompt".
- Will the dataset (M2C) be released? A large-scale dataset of (text, image) pairs would be of great interest to the community.
- The results depicted in Figure 4 are nice, but more should be added (maybe in the appendix) to demonstrate that the method works robustly.
- After how many epochs exactly is the $\mathcal{L}_{FDL}$-loss applied?
- How robust is the model to different choices of $\lambda_1$, $\lambda_2$ and $\lambda_3$ in Eq. (2)?
- What is the effect of codebook size $|\mathcal{Z}|$ on the model?
- Fig. 3 should include a visualization when gradually adding more and more information of the _same_ ground truth prompt, i.e. TC, TC+VC and finally TC+VC+PC.
- How big (if at all) is the problem of overfitting?

__References:__
 - [1]: Jacob Devlin, Ming-Wei Chang, Kenton Lee, and Kristina Toutanova. Bert: Pre-training of deep bidirec-
tional transformers for language understanding. In Proceedings of the Conference on The North American
Chapter of the Association for Computational Linguistics, 2019.
 - [2]: Marjan Ghazvininejad, Omer Levy, Yinhan Liu, and Luke Zettlemoyer. Mask-predict: Parallel decoding
of conditional masked language models. arXiv preprint arXiv:1904.09324, 2019.
 - [3]: Mahajan, Shweta, Iryna Gurevych, and Stefan Roth. "Latent normalizing flows for many-to-many cross-domain mappings." arXiv preprint arXiv:2002.06661 (2020).
 - [4]: Xu, Tao, et al. "Attngan: Fine-grained text to image generation with attentional generative adversarial networks." Proceedings of the IEEE conference on computer vision and pattern recognition. 2018.
 - [5]: Radford, Alec, et al. "Learning transferable visual models from natural language supervision." arXiv preprint arXiv:2103.00020 (2021).


**Time Spent Reviewing:**

10

---

> ### Author Response · Authors · 2021-08-10
> **Author Response to Reviewer wtTT**
>
> We’d like to thank the reviewers for valuable comments.
>
> ------
>
> **Q1. What's the purpose of Figure 1? Neither sketch-to-image nor artistic style transfer are demonstrated with UFC-BERT. This should be clarified in the corresponding caption.**
>
> Figure 1 aims to show three main modalities (TC, VC and PC) of control signals in various tasks and introduces our UFC-BERT can unify flexible multi-modal controls. Though sketch-to-image and artistic style transfer tasks do not be demonstrated in our paper, they can be performed by the UFC-BERT architecture if necessary. Thanks for your suggestion and we will clarify the Figure 1 in revision.
>
> ------
>
>
>  **Q2. It remains somewhat unclear how the cropping for VC and PC works. Is this performed in the latent space of the VQGAN, and correspondences are assigned via the mentioned $\frac{H}{h}\times \frac{W}{w}$ token-pixel correlations? If so, what is the effect of the non-local block (=attention layer) present in the VQGAN? Furthermore, crops can then only be made with a stride of 16 pixels.**
>
> Yes, VC and PC are performed in the latent code space of the VQGAN and correspondences are assigned via the $\frac{H}{h}\times \frac{W}{w}$ token-pixel correlations.
>
> The non-local layer in the VQGAN brings the extra receptive field (outsides the $\frac{H}{h}\times \frac{W}{w}$ block) into each visual code-word, that is, discrete code-words of an image are inherently relevant rather than independent of each other.
> Overall, this characteristic is beneficial to UFC-BERT. First, a code-word will contain extra information about its adjacent code-words, which can help to infer the adjacent code-words. If each code-word is independent, the second stage may be easy to generate inconsistent sequences.
> Second, the extra receptive field provides some holistic information to be utilized, e.g., for clothing generation, the VC for logos also contains some information about the shape and texture, which can improve the holistic consistency of the synthesized clothing.
>
> Furthermore, for crops with an arbitrary shape, we need to select each $\frac{H}{h}\times \frac{W}{w}$ block ($16\times16$ in our experiments) which has any pixel in the required crop. That is, we extend the arbitrary shape to a regular one, consisting of the basic blocks with $\frac{H}{h}\times \frac{W}{w}$ pixels.
> In the future, we will add the images with arbitrary pixel masks into the first-stage training of UFC-BERT, such that we can directly sample the VC and PC in the pixel space and satisfy arbitrary shapes.
>
> ------
>
>
> **Q3. The paper repeatedly states that the relevance and fidelity estimators are "key" to the generation algorithm, but does not provide experiments to justify this claim.**
>
> Actually, our PNAG algorithm is totally dependent on two estimators.
> We justify this claim by the ablation study of the PNAG algorithm in Table 3. The PNAG(w/o. REF) and PNAG(w/o. FDL) discard the relevance estimator and the fidelity estimator respectively, and MNAG is the original Mask-Predict algorithm without any estimator. The results in Table 3 show PNAG outperforms three ablation methods, demonstrating the two estimators effectively utilize the discriminative capability of UFC-BERT and do help improve the synthesis quality.
>
> Moreover, the dynamical termination of PNAG also relies on the scoring of two estimators. The PNAG algorithm will terminate if the comprehensive score does not increase within three consecutive iterations. The dynamical termination can save about 21\% and 29\% inference time for the M2C-Fashion dataset and Multi-Modal CelebA-HQ dataset, respectively. We will add the experiment data in revision.
>
> Besides, Figure 6 shows the scores of relevance and fidelity estimators can guide the inference process towards a better direction and select more realistic images that match the control signals.
>
> ------
>
> **Q4. No metric that explicitly measures correspondence between generated images and corresponding textual descriptions is provided, the paper rather takes a detour by measuring similarity of ground-truth images and samples. However, since there is not a one-to-one correspondence between textual description and image, these metrics (LPIPS, SSIM, PSNR) might not be a good fit. On the contrary, LPIPS has been used to measure diversity of generated samples, where higher scores are better [3]. Using R-precision as in [4] or CLIP-scores [5] could be a simple fix.**
>
> Thanks for your suggestion and we will add the explicit metrics CLIP-score and CLIP-R-precision for correspondence measure between generated images and descriptions in revision.
>
> |Method| CLIP-score|CLIP-R-Precision|
> |:-:|:-:|:-:|
> |VQGAN|0.4863|16.27\%|
> |UFC-BERT|**0.5011**|**17.18\%**|
>
> Concretely, the CLIP-score[5] is the average similarity score of generated images and textual descriptions based on the CLIP model.
> For the M2C-Fashion dataset, we first fine-tune the CLIP model on the training set, and then compute the cosine similarity of image and text features from CLIP. Our UFC-BERT achieves the **0.5011** CLIP-score and outperforms the VQGAN model with the **0.4863** CLIP-score.
>
> As for CLIP-R-precision, we compute the standard R-precision (1 ground truth and 99 randomly selected descriptions) based on the fine-tuned CLIP model instead of the DAMSM model in [4]. This is because the DAMSM model needs to be joinly trained with the main model, which is not suitable for the two-stage architecture. For the M2C-Fashion dataset, our UFC-BERT achieves the **17.18\%** R-precision and outperforms the VQGAN model with **16.27\%** R-precision. We will add the two metrics for two dataset in revision.
>
> ______
>
>
> **Q5. Mixed Comments/Questions/Suggestions**
>
>
> **Q5.1. How does the diversity of samples generated with UFC-BERT compare to an explicit AR-likelihood model?**
>
> By human evaluation, our UFC-BERT can generate more diverse samples compare to the explicit AR-likelihood model. Because the non-autoregressive generation can explore more abundant patterns by the simultaneous sampling at all masked positions, which does not depend on the last token. But the AR-likelihood model often follows the common patterns from the (t-1)-th token to the t-th token and synthesizes a plain image.
>
> Moreover, Figure 4 in the appendix shows the diverse images generated from the same control signals. Concretely, we generate 5 diverse images for given controls, where we select textual controls and the combinations of textual and visual controls. We can observe each generated image complies with multi-modal controls but has obvious differences compared to other images under the same controls.
>
>
> **Q5.2. I would rather call the "counterfactual case" (l. 312 and Fig. 5) a "conflicting prompt".**
>
> Thanks for your suggestion and we will modify the name in revision.
>
>
> **Q5.3. Will the dataset (M2C) be released? A large-scale dataset of (text, image) pairs would be of great interest to the community.**
>
> Of course, we will release the large-scale dataset M2C-Fashion and our code if accepted.
>
> **Q5.4. The results depicted in Figure 4 are nice, but more should be added (maybe in the appendix) to demonstrate that the method works robustly.**
>
> Thanks for your suggestion and we will add more cases as Figure 4 in the appendix. In fact, this method works very robustly.
>
>
>
> **Q5.5. After how many epochs exactly is the $\mathcal{L}_{FDL}$ loss applied?**
>
> We apple the $\mathcal{L}_{FDL}$ loss after 2 epochs for M2C-Fashion and 10 epochs for Multi-Modal CelebA-HQ. This is because the M2C-Fashion dataset is much larger than Multi-Modal CelebA-HQ.
>
>
> **Q5.6. How robust is the model to different choices of ${\lambda}_1$, ${\lambda}_2$ and ${\lambda}_3$  in Eq. (2)?**
>
> The model is very robust to the three hyper-parameters. Because the relevance and fidelity estimation tasks will quickly converge during training and their loss functions will become very low (two orders of magnitude lower than $\mathcal{L}_{MSM}$). That is, the relevance and fidelity estimation tasks will not affect the training of the main task.
>
>
> **Q5.7. What is the effect of codebook size on the model?**
>
> A larger codebook size can provide the stronger expressive ability of the model, but require more data and training resources. We follow the common size 1024 as in VQGAN and can achieve the satisfactory performance.
>
> **Q5.8. Fig. 3 should include a visualization when gradually adding more and more information of the same ground truth prompt, i.e. TC, TC+VC and finally TC+VC+PC.**
>
> Thanks for your suggestion and we will add the gradual visualization in revision.
>
> **Q5.9. How big (if at all) is the problem of overfitting?**
>
> Our UFC-BERT does not have the overfitting problem on M2C-Fashion thank to tens of millions of data. But the model will overfit a bit on the Multi-Modal CelebA-HQ dataset, where $\mathcal{L}_{MSM}$ is about 3.0 for the training set and 4.5 for the validation set. But the overfitting does not affect the synthesized results, where the synthesized images do not exist in the training set. This is because the non-autoregressive generation provide the enough diversity.

---

> > ### Comment · Reviewer_wtTT · 2021-08-22
> > **Thanks for your answers!**
> >
> > Hi,
> >
> > thank you for answering my questions and computing additional metrics. I find that most of the questions have been answered satisfactorily; answering the question of how to solve the cropping "exactly" is something for future work, in my opinion. Only about the diversity I am not so sure yet. What exactly does "human evaluation" mean? One possibility for a quantitative analysis of this is e.g. the paper "Improved Precision and Recall Metric for Assessing Generative Models" (https://arxiv.org/abs/1904.06991).
> >
> > I will keep my score of 7 for now.

---

> > > ### Author Response · Authors · 2021-08-27
> > > **Quantitative Diversity Evaluation**
> > >
> > > Thanks for your suggestion and we add the quantitative diversity evaluation based on the paper  "Improved Precision and Recall Metric for Assessing Generative Models". The paper introduces two metrics Recall and Precision, where the Recall metric assesses the diversity of generated images and the Precision metric evaluates the quality of generated images.
> > >
> > > We compare our UFC-BERT with VQGAN on the two new metrics. Concretely, we use the Inception-v3 as the feature extractor and take activations of the pool_3 layer of the inception-v3 net as image features of generated/real samples. There are two hyper-parameters for the metrics: the number N of evaluated samples and the size K of neighborhoods. We set N to 5,000 and 6,000 (same as the image number in the testing set) for M2C-Fashion and Multi-Modal CelebA-HQ, respectively. And We set K to 3 and 10 for M2C-Fashion and Multi-Modal CelebA-HQ, respectively.
> > >
> > >
> > > |Dataset|Method| Recall |Precision|
> > > | :-:  | :-: | :-: | :-: |
> > > |M2C-Fashion|VQGAN|0.4610 |0.6822|
> > > ||UFC-BERT|**0.4796**|**0.6965**|
> > > |Multi-Modal CelebA-HQ|VQGAN|0.1403 |0.8401|
> > > ||UFC-BERT|**0.1519**|**0.8467**|
> > >
> > > The evaluation results are shown in the above table. We can find our UFC-BERT achieves better diversity (larger Recall) than VQGAN on two datasets. Meanwhile, the image quality (Precision metric) generated from UFC-BERT exceeds those images generated by VQGAN. That is, our UFC-BERT can increase the diversity of generated images while generating high-quality images.
> > >
> > > P.S. the "human evaluation" in our last response means that five volunteers subjectively judge the image diversity by observing the ground truth images and generated images from UFC-BERT and VQGAN.

---

> > > > ### Comment · Reviewer_wtTT · 2021-09-01
> > > > **Diversity Evaluation**
> > > >
> > > > Thanks for providing these additional scores.
> > > > As a last question, how are the sampling hyperparameters for the AR baseline set (i.e. is temperature scaling or nucleus sampling applied?)? Do they affect these results? And finally, what's the reason that $K$ is set differently for M2C-Fashion and MM-CelebA? $K=3$ is the default choice and the authors of https://arxiv.org/abs/1904.06991 found that larger values tend to increase both precision and recall.

---

> > > > > ### Author Response · Authors · 2021-09-02
> > > > > **Hyperparameter Setting**
> > > > >
> > > > > We apply the temperature scaling and top-k sampling for the AR baseline. Concretely, the temperature value is set to 1.2 to improve the diversity of generated images, and the k value in top-k sampling is set to 64 to maintain generation quality. The two hyperparameters will affect the results from the diversity and stability. But our UFC-BERT uses the 1.0 temperature and sets the k value to 200. That is, UFC-BERT does not heavily rely on the sampling hyperparameters and is more robust. Moreover, $K=3$ is the default setting in the paper "Improved Precision and Recall Metric for Assessing Generative Models", but the UFC-BERT and VQGAN both achieve relatively low recall and precision on Multi-Modal CelebA-HQ when $K=3$. Thus, we set $K=10$ to make the comparison more obvious. Note that though $K$ is set to 3, UFC-BERT still outperforms VQGAN on the Multi-Modal CelebA-HQ dataset.

---

> > > > > > ### Comment · Reviewer_wtTT · 2021-09-02
> > > > > > **Sampling Hyperparameters**
> > > > > >
> > > > > > Thank you. If the precision and recall results are to be included in the paper, these hyperparameters should also be reported. If for other evaluations these parameters were adjusted (e.g. FID, CLIP score, ...), this should be stated as well as the sensitivity of the models with respect to these parameters. In addition, the statements made (e.g., "The two hyperparameters affect the diversity and stability results," ...) should be empirically supported. This also applies to statements made earlier in the rebuttal, e.g., "But overfitting does not affect the synthesized results when the synthesized images are not present in the training set." or "The model is very robust to the three hyperparameters."
> > > > > >
> > > > > > To be clear, there is nothing wrong with UFC-BERT not being better than the baseline in all respects, the authors should just say so clearly. I think the NAR generation results are impressive and warrant acceptance.

---

### Official Review · Reviewer_5cvG · 2021-07-16

**Rating:** 6
**Confidence:** 5

**Summary:**

In this paper, instead of using auto-regressive transformer to generate the discrete tokens like "taming transformer", it cleverly employs the well-known BERT model of NLP area, which not only supports the bi-directional conditions for local image editing, meanwhile vastly speeding up the sampling efficiency. The experiments also demonstrate the effectiveness of the proposed method.

**Limitations And Societal Impact:**

Please see the above mentioned comments.

**Main Review:**

Pros:

* The idea of using extra global tokens to judge the current generation status is very interesting.
* The designed progressive non-autoregressive generation is promising.
* This paper is well-written and easy to follow.


Cons:

* What are the details of sampling the re-mask tokens? I am not very clear about the descriptions in Line232-Line233.
* I think the authors ignore one very important problem of the whole framework. The original pixel contents may not be well preserved while giving **the masks with arbitrary shapes**, especially for the boundary regions, which is caused by the mis-alignment of downsampled masks and discrete representation.  This issue should be discussed in the submission.
* The papers which also use bi-directional transformer for image generation like [1,2] should be discussed in the related works.

1. High-Fidelity Pluralistic Image Completion with Transformers, arxiv 2021
2. Diverse Image Inpainting with Bidirectional and Autoregressive Transformers, arxiv 2021

I will consider further raising the scores if the concerns could be well solved.


**Time Spent Reviewing:**

6 hours

---

> ### Author Response · Authors · 2021-08-10
> **Author Response to Reviewer 5cvG**
>
> We’d like to thank the reviewers for valuable comments.
>
> ______
>
> **Q1. What are the details of sampling the re-mask tokens? I am not very clear about the descriptions in Line232-Line233?**
>
> We are sorry for the unclear descriptions of sampling the re-masked tokens in Line232-Line233. Our Mask Step iteratively re-masks a subset of tokens with low probability scores. We modify Line232-Line234 as follows:
>
>
> > Specifically, we re-mask $n$ tokens of ${\bf I}^{(t-1, out)}$ to produce each ${\bf I}^{(t, in)}_{b}$. We first sample $N_I-n$ tokens from a multinomial distribution ${Pr}^{(t,in)}$ proportional to the probability scores ${\bf Y}^{(t-1)} = ({Y}^{(t-1)}_1, {Y}^{(t-1)}_2, \dots, {Y}^{(t-1)}_N)$, computed by ${Pr}^{(t,in)} = Softmax({\bf Y}^{(t-1)})$.
> > And other tokens are re-masked and re-predicted at the next Predict Step. Here $n = N_I \cdot (\beta + \frac{T-t}{T-1}\cdot({\alpha}-\beta))$, where $\alpha$ is the initial mask ratio, $\beta$ is the minimum mask ratio, and $T$ is the maximum possible number of iterations, such that the number of tokens to re-mask gradually decreases after every iteration.
>
> Note that the formula of the re-masked number $n$ is also corrected.
>
> ______
>
>
> **Q2. I think the authors ignore one very important problem of the whole framework. The original pixel contents may not be well preserved while giving the masks with arbitrary shapes, especially for the boundary regions, which is caused by the mis-alignment of downsampled masks and discrete representation. This issue should be discussed in the submission.**
>
> Thanks for your comment and the mis-alignment of sampled masks and discrete representation is an important issue for the two-stage architecture. Actually, we sample the visual controls (VC) and preservation controls (PC) in the quantized code space, hence we can only crop or mask a regular region of the original image, which consists of the basic blocks with $\frac{H}{h}\times \frac{W}{w}$ pixels ($16\times16$ in our experiments). For arbitrary shape with irregular boundaries, we will select each basic block which has any pixel in the required shape. That is, we extend the arbitrary shape to a regular one with redundant pixels.
> But this is just a sub-optimal strategy and cannot completely solve the mis-alignment problem. In the future, we will add the images with arbitrary pixel masks into the first-stage training of the two-stage architecture, such that we can directly sample the VC and PC in the pixel space and avoid the problem.
>
>
> ______
>
>
> **Q3. The papers which also use bi-directional transformer for image generation like [1,2] should be discussed in the related works.**
>
> Thanks for your suggestion and we will discuss the related works [1,2] in revision. The two works also adopt the two-stage architecture for image completion. They first develop a bi-directional transformer to recover low-resolution image structures in the discrete space, and then synthesize the high-resolution image by a CNN-based upsampling model. There are many differences between [1,2] and our work:
>
>
> - We focus on conditional image synthesis with universal multi-modal controls, but [1,2] only tackle the image completion task (i.e. PC in our paper) and cannot be simply extended to other tasks.
> - We employ the non-autoregressive generation with BERT and further design the PNAG algorithm to improve the relevance and fidelity of synthesized images. But [1,2] still employ the sequential (left-to-right) generation by the Gibbs sampling[1] or permutation strategy[2] like in XLNet[3]. That is, though [1,2] introduce the bi-directional context during generation, they cannot drastically improve the inference speed like us.
> - We use the design of VQGAN for image quantization and can preserve image information as much as possible. However, [1,2] directly downsample the input image with missing pixels and discretize the downsampled image by k-means clustering of RGB pixel values from ImageNet dataset. The process inevitably causes the severe loss of image information.
>
>
> 1. High-Fidelity Pluralistic Image Completion with Transformers, arxiv 2021
> 2. Diverse Image Inpainting with Bidirectional and Autoregressive Transformers, arxiv 2021
> 3. XLNet: Generalized autoregressive pretraining for language understanding, NeurIPS 2019

---

### Decision · Program_Chairs · 2021-09-27

**Decision:**

Accept (Poster)

**Comment:**

The paper proposed a method to unify any number of multi-modal control signal inputs for conditional image synthesis. The key thing was a transformer model that could convert a variable number of multi-modal control inputs to the discrete latent space of a VQGAN. All the reviewers rated the paper above the bar, with one reviewer upgraded the score from 6 to 7 after the rebuttal. The meta-reviewer agreed with the assessment and concluded the paper is above the bar. Please include the reviewer feedback in the updated manuscript in the final version.